# Conformational dynamics of the ABC transporter McjD seen by single-molecule FRET

Florence Husada[1,†], Kiran Bountra[2,3,†], Konstantinos Tassis[1], Marijn de Boer[1], Maria Romano[2,3], Sylvie Rebuffat[4], Konstantinos Beis[2,3,*] & Thorben Cordes[1,5,**]

## Abstract

ABC transporters utilize ATP for export processes to provide cellular resistance against toxins, antibiotics, and harmful metabolites in eukaryotes and prokaryotes. Based on static structure snapshots, it is believed that they use an alternating access mechanism, which couples conformational changes to ATP binding (outward-open conformation) and hydrolysis (inward-open) for unidirectional transport driven by ATP. Here, we analyzed the conformational states and dynamics of the antibacterial peptide exporter McjD from *Escherichia coli* using single-molecule Förster resonance energy transfer (smFRET). For the first time, we established smFRET for an ABC exporter in a native-like lipid environment and directly monitor conformational dynamics in both the transmembrane- (TMD) and nucleotide-binding domains (NBD). With this, we unravel the ligand dependences that drive conformational changes in both domains. Furthermore, we observe intrinsic conformational dynamics in the absence of ATP and ligand in the NBDs. ATP binding and hydrolysis on the other hand can be observed via NBD conformational dynamics. We believe that the progress made here in combination with future studies will facilitate full understanding of ABC transport cycles.

**Keywords** ABC transporter; conformational dynamics; FRET; single-molecule studies

**Subject Categories** Membrane & Intracellular Transport; Structural Biology

The EMBO Journal (2018) 37: e100056

## Introduction

ABC transporters are essential membrane proteins found in both eukaryotic and bacterial cells that facilitate the uphill transport of ions and chemically diverse compounds in an ATP-dependent manner (Holland *et al*, 2003). They are involved in numerous cellular processes including nutrient import, metal homeostasis, detoxification, and antigen processing. The ABC exporter family plays a major role for the extrusion of toxic compounds and has relevance for human diseases, tumor resistance, and bacterial virulence. All ABC transporters share a common architecture consisting of a transmembrane domain (TMD) for substrate recognition and transport, and a nucleotide-binding domain (NBD) that converts the chemical energy of ATP into conformational changes for transport (Beis, 2015). The structures of several homodimeric (Dawson & Locher, 2006; Ward *et al*, 2007; Perez *et al*, 2015) and heterodimeric ABC transporters (Hohl *et al*, 2012; Noll *et al*, 2017) revealed distinct conformations and suggest, in combination with biophysical studies (e.g., EPR and NMR; Dong *et al*, 2005; Zou *et al*, 2009; Bountra *et al*, 2017; Timachi *et al*, 2017; Barth *et al*, 2018), that they undergo large conformational changes during transport. Their complex architecture is, however, a fundamental hurdle to fully understand the coupling between conformational changes, substrate binding, ATP binding and hydrolysis, and transport. Such detailed mechanistic models of transport would require not only the knowledge of conformational states from static snapshots, but also their interconversion dynamics.

Based on the available structural, biophysical, and biochemical data for exporters, two transport mechanisms have been proposed, the alternating access mechanism (Dawson & Locher, 2006; Ward *et al*, 2007; Choudhury *et al*, 2014; Bountra *et al*, 2017) and the outward-only mechanism (Perez *et al*, 2015). ABC exporters that use the alternating access mechanism switch between inward- and outward-facing states with transmembrane helices intertwining, which exposes the ligand binding site alternatively to the inside or outside of the membrane, a process that is believed to be coupled to ATP binding to hydrolysis. The outward-only mechanism was proposed in light of the structural and functional data of the lipid-linked oligosaccharide flippase PglK from *Campylobacter jejuni* (Perez *et al*, 2015); here the inward-facing conformation is not

1  Molecular Microscopy Research Group, Zernike Institute for Advanced Materials, University of Groningen, Groningen, The Netherlands
2  Department of Life Sciences, Imperial College London, London, UK
3  Rutherford Appleton Laboratory, Research Complex at Harwell, Didcot, UK
4  Communication Molecules and Adaptation of Microorganisms Laboratory, (MCAM, UMR 7245 CNRS-MNHN), Muséum National d'Histoire Naturelle, Centre National de la Recherche Scientifique, Sorbonne Universités, Paris, France
5  Physical and Synthetic Biology, Faculty of Biology, Ludwig-Maximilians-Universität München, Planegg-Martinsried, Germany
   *Corresponding author. Tel: +44 1235 567809; E-mail: kbeis@imperial.ac.uk
   **Corresponding author. Tel: +49 89 2180 74623; E-mail: cordes@bio.lmu.de
   †These authors contributed equally to this work

required for substrate translocation since it can be directly recruited from the membrane and ATP hydrolysis at the NBDs is transmitted to the TMD to drive substrate release.

ABC transporters that utilize the alternating access mechanism undergo large conformational changes during the transport cycle suggested by crystal structures (Ward *et al*, 2007) and EPR-based Double Electron Electron Resonance (DEER) measurements (Dong *et al*, 2005; Borbat *et al*, 2007; Mishra *et al*, 2014) of the lipid A flippase MsbA from *Escherichia coli*. Although the DEER measurements proposed that ATP triggers the opening of the TMD to the periplasmic side for release of the substrate, it is unclear how precisely these changes are coupled to ATP binding and hydrolysis. It has been proposed that only homodimeric transporters employ this mechanism. Opening of the TMD of heterodimeric ABC transporters is coupled to ATP hydrolysis rather than binding due to the nature of their NBDs that contain a consensus and degenerate site for ATP hydrolysis (Mishra *et al*, 2014; Timachi *et al*, 2017; Barth *et al*, 2018). In contrary, we have previously shown that the antibacterial peptide ABC transporter McjD, which confers bacterial cells with self-immunity against the antibacterial peptide MccJ25, adopts distinct occluded conformations (Choudhury *et al*, 2014; Bountra *et al*, 2017) to other bacterial exporters including MsbA (Ward *et al*, 2007). The TMD is occluded to either side of the membrane in the presence or absence of nucleotides that was also supported by Pulsed Electron-Electron Double Resonance (PELDOR) measurements in bicelles (Bountra *et al*, 2017). Occluded conformations have also been reported for other ABC transporters (Lin *et al*, 2015; Perez *et al*, 2015; Morgan *et al*, 2017). We have biochemically shown that McjD adopts a transient outward-open conformation that it is probably not long-lived or well populated to be trapped by PELDOR (Bountra *et al*, 2017). Although both MsbA and McjD utilize the alternating access mechanism for export of their substrate, the overall mechanism is not fully conserved and requires further characterization.

So far, however, all the conformations that have been reported for ABC transporters are static and do not provide insights into the conformational changes and dynamics that govern the movement of NBDs upon ATP binding and hydrolysis or that of the TMDs during substrate transport across the membrane bilayer. In recent years, it has, however, become apparent that transporters show conformational dynamics and that those have to be understood and characterized (Locher, 2016). For this, single-molecule Förster resonance energy transfer (smFRET; Lerner *et al*, 2018) has been used to monitor these conformational dynamics of transporters, e.g., the P-type $Ca^{2+}$-ATPase (LMCA1) from *Listeria monocytogenes* (Dyla *et al*, 2017), secondary transporters such as the aspartate/$Na^+$ symporter from *Pyrococcus horikoshii* (Akyuz *et al*, 2013), and the leucine/$Na^+$ symporter LeuT from *Aquifex aeolicus* (Zhao *et al*, 2010). While there are also smFRET studies of ABC importers regarding the conformational dynamics of the substrate binding domains (SBDs; Gouridis *et al*, 2015) or interactions of SBD-TMD (Goudsmits *et al*, 2017a; Yang *et al*, 2018), there are no studies of conformational dynamics and crosstalk between TMDs and NBDs in ABC exporters in native-like lipid environment (Verhalen *et al*, 2012).

Here, we have analyzed the antibacterial peptide exporter McjD from *E. coli* in proteoliposomes using smFRET via labeling of specific residues in the TMD and NBDs to monitor the conformational dynamics during the transport cycle. The developed assay has the potential to provide full understanding of the transport mechanism and can be used for both ABC importers and exporters. In this study, it was used to answer the following mechanistic questions regarding McjD: (i) How is ATP binding or hydrolysis coupled to the opening of the TMD? In McjD, crystal structures and PELDOR data in the presence of the ATP-analogue adenosine 5′-(β,γ-imido) triphosphate (AMPPNP), mimicking ATP binding, and ADP-vanadate (mimicking ATP hydrolysis) could not reveal an open cavity (Choudhury *et al*, 2014; Bountra *et al*, 2017). (ii) What are the intrinsic and ligand-induced conformational dynamics of McjD as a model system for ABC exporters and on what timescales do they occur?

To answer these questions, we have studied McjD reconstituted in liposomes under equilibrium conditions to obtain a picture of the stable conformational states and non-equilibrium conditions to understand how conformational changes are triggered by substrates using smFRET. We demonstrate that the NBDs have intrinsic conformational dynamics in their apo state on the 100 ms timescale, while the TMDs remain static. ATP binding and hydrolysis on the other hand take much longer and it is observed via NBD conformational dynamics on the timescale of several seconds, which is a value compatible with reported biochemical data on McjD. Our assays also show that the TMD remains occluded in the presence of nucleotides and only opens when both the substrate MccJ25 and ATP are present. Our study represents major advances in using smFRET for mechanistic studies of ABC transporters in a native-like lipid environment. Furthermore, this is the first report showing that an ABC exporter requires both ATP and substrate binding to drive the formation of the outward-open conformation. Thus, we propose that opening of the McjD cavity is tightly coupled to both ATP and MccJ25 binding and present a refined model for the transport cycle.

## Results

### Single-molecule FRET monitors the conformational states of the ABC exporter McjD

smFRET has emerged as a tool for investigating conformational dynamics of biomacromolecules over the past 20 years (Lerner *et al*, 2018). This approach relies on mapping a relevant reaction coordinate, i.e., for ABC transporters spatial separation of characteristic residues between TMD and NBD domains, and provides "low-resolution" structural information in the form of a distance. On the other hand, however, it is a very powerful technique to measure real-time conformational changes and dynamics unlike other biophysical techniques such as EPR, where the sample has to be frozen. In this paper, we have established smFRET to directly monitor the conformational states of ABC transporters. Our assay is tailored to monitor conformational states and dynamics during substrate export and relies on the use of single cysteines at the periplasmic side of the TMD, Y64C, and L67C (Fig 1A) and the cytosolic NBDs, native C547, of McjD (Fig 1A); all single mutants were introduced to cys-less McjD (via mutation of native cysteines to serine).

It can be clearly seen that, due to the homodimeric nature of McjD, introduction of one cysteine results in the possibility to label the protein with two fluorophores. This fulfills the requirement of

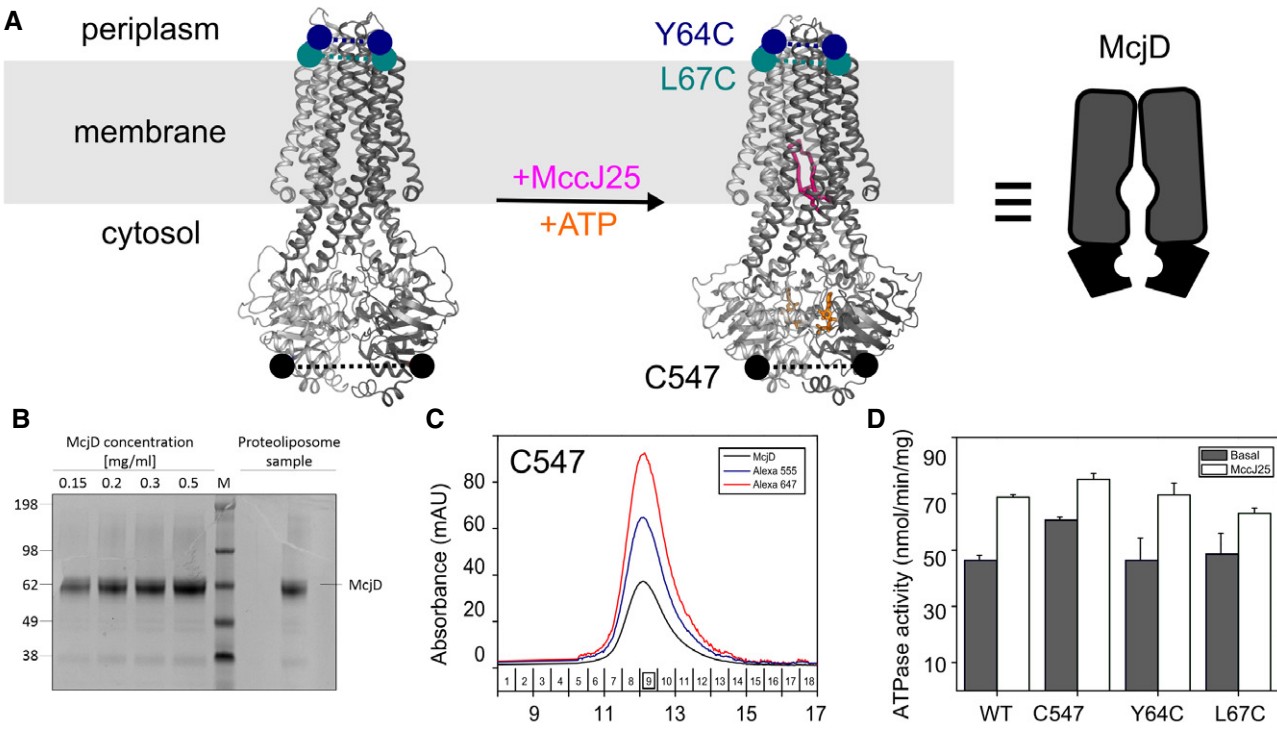

**Figure 1.  Biochemical characterization of McjD and smFRET assay design.**

A   Schematic representation of the smFRET assay using crystal structure snapshots of McjD. In the crystal structure, nucleotides and ATP-Mg are in orange sticks, and the peptide MccJ25 is in pink. Each half transporter is colored in dark and light gray. McjD is shown in a view along the plane of the membrane which is depicted in light gray. Colored balls show the position of the cysteines used for probing conformational changes. PBD codes for McjD in inward-occluded conformation are 5OFP (left; Bountra *et al*, 2017) and in the outward-occluded conformation are 4PL0 (right; Choudhury *et al*, 2014).

B   Coomassie-stained SDS–PAGE of McjD-C547 reconstituted in liposomes. The amount reconstituted was evaluated against detergent-purified protein.

C   Chromatogram of McjD-C547 (black) labeled with Alexa 555 (blue) and Alexa 647 (red)-maleimide fluorophore using a Superdex 200 size-exclusion column. The fraction containing the highest degree of labeling is marked by a square.

D   The ATPase activity of all labeled variants C547, Y64C, and L67C was determined in liposomes and compared to wild-type, wt. All labeled McjD variants show basal- and ligand-induced ATPase activities in liposomes comparable to that of wild-type protein, indicating that labeling does not interfere with their activity. Error bars were calculated from two replicates from two independent reconstitutions (mean ± standard deviation, *n* = 2). The ruler character of FRET using the fluorophore pair Alexa555/Cy5 was supported by data of dsDNA in Fig EV1B, where different base-pair distances between both dyes correlate directly with the apparent FRET efficiency.

FRET to introduce both donor and acceptor fluorophore. Furthermore, structural analysis suggests that the apo conformation of McjD has distinct distances between the two labeled positions compared to the nucleotide-bound conformation for McjD C547 (but not for Y64C and L67C), a fact that should allow assessment of the biochemical and conformational states of McjD via smFRET. The FRET efficiency changes are related to the distance of the respective variants and thus indicate the changes that occur upon addition of ATP or the substrate MccJ25. For example, the crystal structure estimates that the nucleotide-free form of McjD should show a larger NBD separation (low FRET), while this should be shorter in the nucleotide-bound form (high FRET; Choudhury *et al*, 2014; Bountra *et al*, 2017).

McjD was purified in detergent and reconstituted into proteoliposomes as previously described (Fig 1B; Choudhury *et al*, 2014; Bountra *et al*, 2017). For smFRET, McjD and variants were labeled stochastically with Alexa555 and Alexa647 (see "Materials and Methods"). The gel-filtration purification of donor–acceptor-labeled McjD is shown in Fig 1C. The detergent-solubilized protein runs as a mono-disperse peak with significant absorption at both the donor

and acceptor wavelengths at 532 and 640 nm, respectively (Fig 1C). The labeling efficiency of the two fluorophores for McjD-C547 was determined to be ~60%. All variants were biochemically characterized using a previously published ATPase assay (Choudhury *et al*, 2014; Bountra *et al*, 2017). Neither the introduction of cysteines nor the addition of fluorescent labels affected the activity of the transporter in proteoliposomes; this is indicated by similar basal and ligand-induced ATPase activity compared to wild-type McjD (Fig 1D). All interpretations given below are supported by comparison of the observed E* values of McjD with respect to static dsDNA ruler structures (Fig EV1A and B) that we have previously described (Ploetz *et al*, 2016).

We first assessed the conformational states of McjD under equilibrium conditions in the detergent-solubilized state. We used three different labeling positions in McjD to monitor the conformational states of the periplasmic TMD region and the cytosolic ATPase domain (Figs 1A and 2A). Since detergent-solubilized McjD diffuses freely in solution, we used confocal microscopy with alternating laser excitation (ALEX) to relate FRET efficiency values of single McjD molecules to their respective conformational states

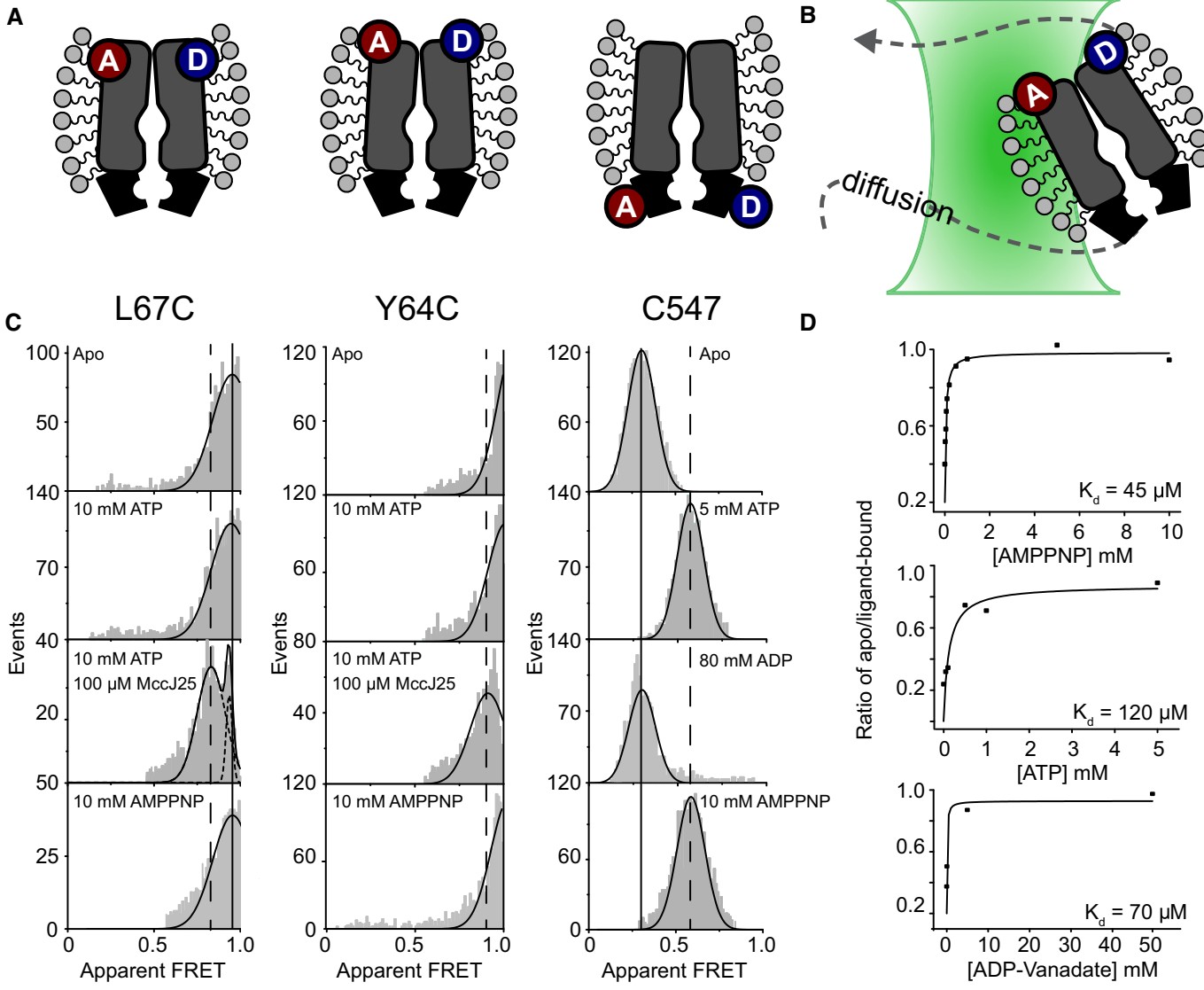

**Figure 2. Freely diffusing McjD in detergent studied by smFRET.**

A, B Schematic overview of McjD labeling schemes (A) and diffusion-based experiments (B) using confocal microscopy. Stochastic labeling is performed on the NBD (native cysteine C547) and TMD mutants (Y64C; L67C).

C Confocal single-molecule analysis with ALEX of labeled McjD in detergent under different conditions as indicated.

D Apparent ligand binding affinity values of McjD for AMPPNP (top), ATP (middle), and ADP-vanadate (bottom). Data points were obtained from the areas of fits from the E* histograms considering the ratio ligand-free/(ligand-free + ligand-bound) at the indicated substrate concentrations. Extraction of $K_d$ values from a fit to the data is described in the Materials and Methods section. Data for detergent-solubilized McjD for additional biochemical conditions are shown in Fig EV2.

(Fig 2B). From the available crystal structures, we extracted $C_a$-$C_a$ distances between the two labeling positions. It is worthwhile to note that, in our smFRET assays, we do not determine absolute distances, but only use relative distance changes via the setup-dependent apparent FRET efficiency E*. The qualitative relation between E* and the $C_a$-$C_a$ distances do, however, allow us to assess whether FRET experiments relate well to distances expected from the crystal structures. For the periplasmic TMD mutants, Y64C and L67C, the crystal structures predict $C_a$-$C_a$ distances of 2.6 and 2.9 nm, respectively (without taking any fluorophore properties into account). The Förster radius of Alexa555/647 is $R_0$ = 5.1 nm. At this distance, FRET efficiency is theoretically expected to be 0.5; interprobe distances above this value are thus expected to show smaller values of E*, while shorter distances are expected to show larger E* values. Since $C_a$-$C_a$ distances of both TMD mutants are shorter than $R_0$, we expect high-FRET efficiency values significantly above 0.5. The data are coherent with this expectation, based on crystal structure predictions as seen from Fig 2C. FRET distribution of apo-McjD center is around 0.95 for L67C and ~1.07 for Y64C (see Table 1 and Fig EV2 for mean FRET values). The distance extracted for C547, located at the NBDs, from the crystal structure of the apo-McjD is 4.8 nm, in agreement with the mean observed FRET efficiency of apo-McjD of ~0.31 when also considering added flexibility of the dyes by their linkers.

**Table 1.  Mean apparent FRET values from a 1D Gaussian fit.**

| | Detergent | | | Liposomes | | |
|---|---|---|---|---|---|---|
| | L67C | Y64C | C547 | L67C | Y64C | C547 |
| Apo | 0.95 | 1.07 | 0.31 | 0.69 | 0.81 & 0.60 | 0.32 |
| ATP | 0.95 | 1.01 | 0.56 | n.d. | n.d. | 0.495 |
| ATP + Vanadate | n.d. | n.d. | 0.56 | n.d. | n.d. | n.d. |
| ATP + MccJ25 | 0.83 | 0.91 | n.d. | 0.51 | 0.82 & 0.52 | n.d. |
| ADP | 0.95 | 0.99 | 0.31 | n.d. | n.d. | n.d. |
| ADP + Vanadate | 0.95 | 0.99 | 0.30 & 0.56 | n.d. | n.d. | n.d. |
| Vanadate | n.d. | n.d. | 0.31 | n.d. | n.d. | n.d. |
| AMPPNP | 0.95 | 1.03 | 0.54 | n.d. | n.d. | n.d. |

The table provides an overview of mean apparent FRET values $E^*$ (setup dependent) for different environmental conditions. It has to be noted that values in detergent were derived from a different microscope setup than for liposomes and can thus not be compared directly; n.d., not determined.

Having established that our smFRET assays are coherent with structural predictions, we tested conditions that mimic relevant stages of the ABC transport cycle. From diffusion-based experiments, these show the statistically relevant states and related conformations of McjD for the following: (i) ATP binding, hydrolysis, and turnover with/without substrate MccJ25; (ii) the pre-hydrolysis state using the ATP-analogue AMPPNP; (iii) the transition state during ATP hydrolysis with ATP-Vanadate; and finally (iv) the post-hydrolysis state using either ADP or ADP-vanadate (Figs 2 and EV2).

For both TMD mutants, the addition of ATP, its non-hydrolyzable analogue AMPPNP, or transition-state mimic such as ATP-vanadate did not alter their conformational states (Fig 2C). Interestingly, the addition of MccJ25 and ATP triggered opening of the periplasmic gate as observed by a small but significant shift toward lower FRET efficiency values (Fig 2C). Strikingly, the observation of opening occurred under equilibrium conditions and indicates that the outward-open conformation is visited by McjD during the transport cycle and can be triggered by both ATP and the MccJ25 peptide. We observe a slightly smaller opening of the TMDs in the absence of nucleotides but the presence of MccJ25 that suggests the peptide can induce an open TMD during its rebinding (Fig EV2B). We have shown, however, that McjD transports MccJ25 in an ATP-dependent manner. We explain the observations by the fact that ABC exporters can transport their substrates bidirectionally, and this previously described effect manifests in a structural reorientation of the TMD and opening that is similar to that observed during transport with MccJ25 and ATP but only occurs at high concentrations of peptide via rebinding (Grossmann *et al*, 2014). The ABC transporter associated with antigen processing (TAP) has been shown to be unidirectional despite observation of rebinding of peptide, which is essential for its physiological function (Grossmann *et al*, 2014).

On the contrary, the NBDs were very sensitive to ATP (and analogues) and McjD-C547 showed pronounced conformational changes upon ATP, AMPPNP, ATP-vanadate, and ADP-vanadate addition (Fig 2C). Whenever ATP is already present, MccJ25 did not alter the observed conformational state any further. The only conditions that were incompatible with conformational change are sole addition of either peptide, ADP, or vanadate alone (Tables 1 and EV1, Fig EV2).

The conformational changes in McjD-C547 occur with apparent affinities in the 50–100 μM range for ATP, AMPPNP, and ADP-vanadate (Fig 2D), matching reported values for binding affinities of other ABC transporters (Siarheyeva & Sharom, 2009). The data show that binding of the nucleotides (ATP, AMPPNP, and ADP-vanadate) at the NBDs triggers conformational changes into their closed dimer form (Fig 2C). It is only after ATP hydrolysis (ADP-vanadate, Fig EV2; and ADP, Fig 2) that reopening of the NBDs is seen, which we believe is linked to a reset of the transporter.

Altogether, the data based on detergent-solubilized McjD suggest an alternating access mechanism, where conformational changes are triggered by ATP (engagement of the NBDs from inward-open to outward-occluded) and substrate binding (opening of the TMD to outward-open). ATP hydrolysis seems to only reset the transporter since ATP binding is coupled to conformational changes in the NBDs and TMD. This interpretation is further supported by the fact that ADP binding does not induce conformational changes. The data also suggest tight coupling between ATP and MccJ25 binding and subsequent substrate transport. Since the detergent environment is, however, not physiologically relevant, we further verified these results (and their interpretations) with surface-immobilized liposome-reconstituted McjD.

## McjD displays similar conformational states and changes in detergent and proteoliposomes

For studies of surface-immobilized liposomes containing one transporter, we used a low protein-to-lipid ratio of ~1:1,000 (w/w). To verify proper reconstitution, we observed the time duration of fluorescence bursts in a confocal microscope with McjD in detergent and in proteoliposomes and found the expected increase for proteoliposomes to about 10–20 ms (Fig EV3A) compared to ~1 ms for McjD in detergent micelles. These values are similar to reported ones for other diffusing liposome systems (Seo *et al*, 2014). Finally, we verified random insertion (inside-out and inside-in) of McjD into the proteoliposomes with a ratio of ca. 60:40 (see Materials and Methods and Fig EV3B).

McjD reconstituted in liposomes was immobilized in custom-made flow cells on PEGylated coverslips via liposomes containing biotin-DOPE (Fig 3A). Confocal scanning microscopy was

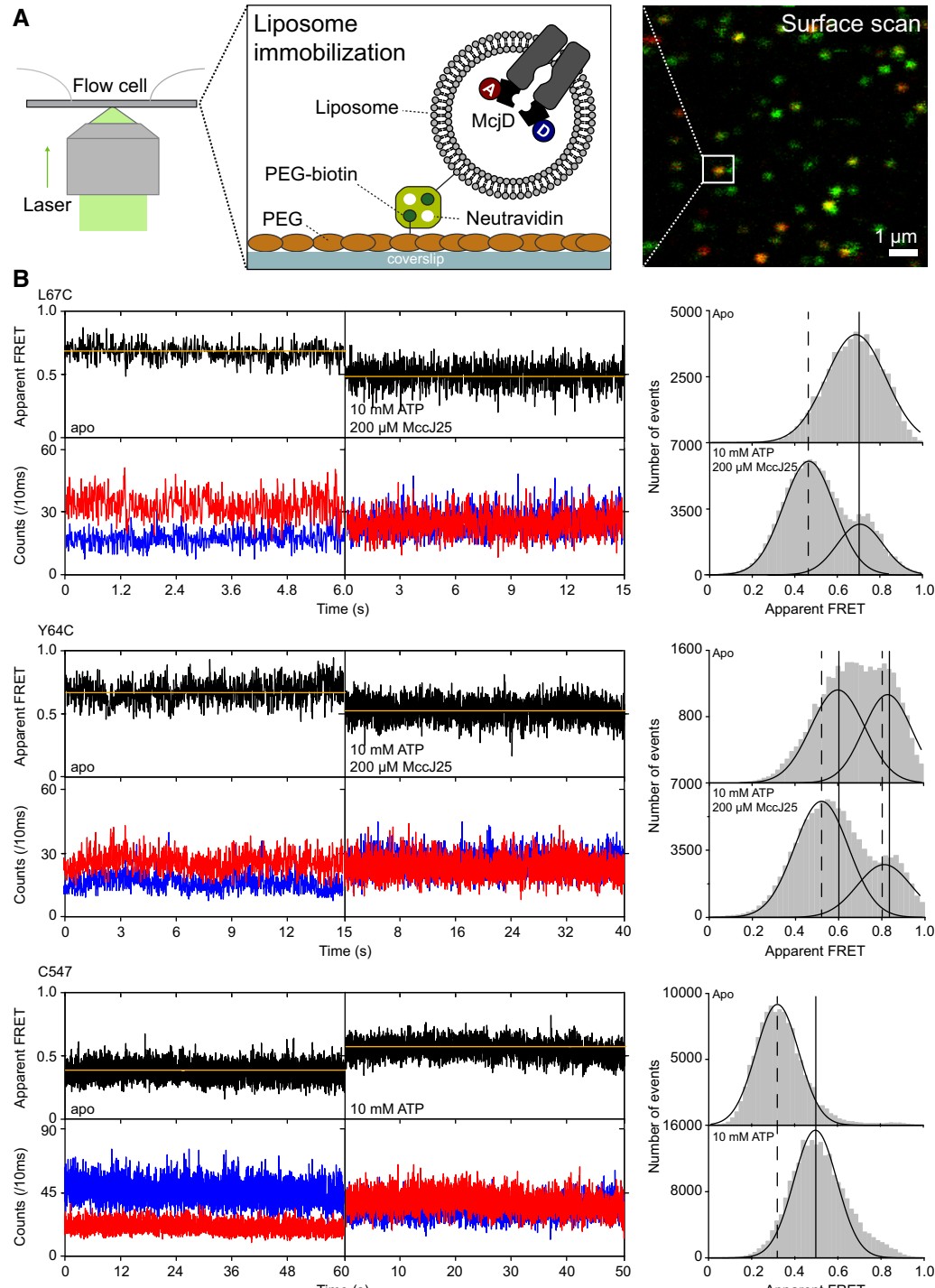

**Figure 3. Conformational states of surface-immobilized McjD in proteoliposomes.**

A  Schematic showing the immobilization of McjD to the surface: Labeled McjD reconstituted in biotinylated DOPE lipid was immobilized to a PEG-biotin-coated surface in a flow cell via a neutravidin-tag on the proteoliposome. A typical surface scan is shown in false-color representation on the right (orange, double-labeled McjD; green, McjD with donor fluorophore only; red, McjD with acceptor fluorophore only).

B  Representative fluorescence time traces (blue, donor signal; red, acceptor signal; black, FRET signal; yellow, fit) of McjD labeled with Alexa555 and Alexa647-maleimide with indicated concentrations of corresponding substrate: McjD L67C with 10 mM ATP and 200 μM MccJ25, McjD Y64C with 10 mM ATP and 200 μM MccJ25, and McjD C547 with 10 mM ATP. The respective right panels are the projections of accumulated time traces in the absence and presence of substrate for each variant. We note that differences between the setup-dependent apparent FRET values are not influencing our interpretations since only relative changes of FRET efficiency are interpreted. Experimental support for proper reconstitution is provided in Fig EV3A showing longer bursts and distinct TEV cleavage for liposomes compared to detergent.

performed as described before (Gouridis *et al*, 2015), and single liposomes could be identified with $10 \times 10$ μm search areas (Fig 3A). Individual McjD transporters with donor and acceptor dye were identified manually. In the data evaluation process, we discarded events with donor- or acceptor-only character or where more than one bleaching step was seen in either detection channel. Using this procedure, we observed fluorescence transients with a time resolution of 10 ms and observation times of multiple seconds [Fig 3B, donor = blue, acceptor = red, FRET = black, orange = most probable state trajectory of Hidden Markov Model (HMM); see Materials and Methods]. FRET histograms were obtained by projection of all E* values from multiple traces ($N > 50$) for each condition; see Fig 3B, right.

The observed FRET efficiency values for apo-McjD in proteoliposomes are fully consistent but not identical with our detergent data. Both TMD mutants show high E* values, mean of 0.69 (L67C) and 0.81 (Y64C), Table 1 and Fig EV2. The differences between both mutants, not the absolute FRET values, are consistent with the detergent indicating that the FRET assay correctly monitors TMD conformation in the liposomes. Also, the conformational changes triggered by addition of peptide and ATP shift the values toward of 0.52 for both mutants (Fig 3B, L67C, Y64C). The fact that the FRET changes are more pronounced in the liposomes represents the increased sensitivity of the FRET assay since the larger absolute distances for both TMD mutants in the liposomes make FRET more sensitive (FRET has the greatest sensitivity for structural changes around the Förster radius). In summary, the TMD shows opening in the presence of both ATP and MccJ25. Considering the absolute values of apparent FRET seems to indicate overall larger opening of the TMD in liposomes compared to detergent environment (Table 1).

The cumulative FRET histograms for both TMD mutants also comply with the fact that only 60% of McjD transporters have the correct orientation (cytosolic NBD facing the outside of the liposome) to react to addition of MccJ25 and ATP, while 40% all of molecules stay in the apo state (Fig 3B). We observed a bimodal distribution of the Y64C in its apo state which we associate to be more dynamic since it is located further up TM1, compared to L67C, and it is likely be less restricted in motion by the proteoliposome's lipid bilayer, thus showing a broader distribution. This will not manifest itself in detergent experiments since here the overall distances are shorter and thus less ideal for the dynamic range of FRET to capture it. The NBD FRET data for C547-McjD in liposomes are almost identical with the detergent ones and show low-FRET apo molecules shifting toward a high-FRET ATP-bound state (Table 1 and Fig EV2). The only minor difference concerns the FRET efficiency of the ATP-bound state which seems slightly more expanded in the liposomes.

We also found that the random orientation of McjD in liposomes is less clearly visible in Fig 3B for C547 compared to both TMD mutants. We see the origin of this in altered photophysical properties of the dyes. For C547, there is a significant fraction of molecules (30–50%) that display very short photobleaching lifetimes < 1 s, in contrast to long-lived molecules similar to those shown in Fig 3B. All short-lived molecules show low-FRET (apo) character independently of the presence or absence of ATP. This tells us that whenever the transporter is labeled at the NBDs it requires photostabilizer in the buffer to obtain long FRET traces. When the

fluorophores reside on the TMDs, however, these intrinsically promote higher photostability and more uniform traces from both transporter orientations are observed. As a consequence, the TMD populations (inside-out and outside-in) are equally weighted in the cumulative FRET histograms, while for C547 the photostable (NBD-out) population dominates the cumulative histograms over the shorter traces from the NBD-in orientation that cannot react on addition of ATP.

Altogether, the combined data of detergent-solubilized and liposome-reconstituted McjD are not identical but consistent—a finding that is interesting in itself. This observation is consistent with previous reports that MsbA displays different degrees of domain separation in micelles and lipid bilayers (Zoghbi *et al*, 2016). We have shown that lipids modulate the activity of McjD (Mehmood *et al*, 2016) and the non-identical FRET values in this study between the detergent and liposome data provide further evidence on the role of lipids on modulating the transport activity of ABC transporters. Furthermore, the data remain fully compatible with the alternating access mechanism, where conformational changes are triggered by ATP (closing of NBDs from inward-open) and substrate binding (opening of the TMD for outward-open). The data also suggest tight coupling between ATP and MccJ25 binding and subsequent substrate transport and suggest that hydrolysis is required for resetting the transporter.

### The NBDs of McjD show conformational dynamics in the absence of ligands, while the TMDs are static

Next, we used smFRET to investigate intrinsic conformational dynamics of McjD, i.e., in the absence of any ligand or nucleotide. For this, we used a time resolution and data binning of 10 ms. For both TMD mutants, we did not observe any intrinsic FRET changes that go beyond experimental noise, when analyzing data sets with $N > 100$ molecules. Both TMD mutants show exclusively traces similar to those shown in Fig 3B (L64C, Y67C, apo). These data suggest that the TMD shows no detectable conformational changes on timescales > 10 ms in the absence of substrate and ATP.

In contrast, a significant fraction of 15% of all McjD-C547 FRET traces showed short stepwise changes of FRET efficiency from the lower apo level of 0.3 to values > 0.5, similar to those found in the presence of ATP (Fig 4A, apo vs. ATP-bound). The dwells in the high-FRET state are short, and the transitions in FRET efficiency were accompanied by anticorrelated change of donor and acceptor fluorescence intensity. These transitions occur with low frequency and on average only once per trace (Fig 4B). A histogram of all observed transition dwells revealed an average lifetime of $82 \pm 25$ ms of the dimerized ATP-free NBD state from an exponential fit of the dwell-time distribution (Fig 4C). Finally, we also verified ATP-induced NBD switching with detergent-solubilized McjD on the surface to exclude unwanted influence on McjD due to surface immobilization. Surface-immobilized McjD-C547 and free-diffusing molecules show a high degree of consistency (compare Fig 2 with Fig EV4) supporting the claims made here.

These data support the interpretation of intrinsic conformational flexibility of the NBD domains but high structural rigidity in the TMD in the absence of ATP and MccJ25. The flexibility of the NBD domains might also be the basis for the observed basal ATPase activity of McjD and other ABC transporters. The findings also show that

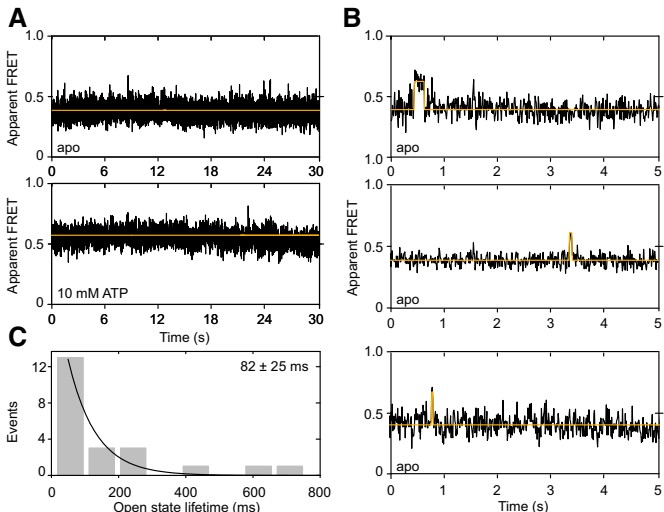

**Figure 4. Intrinsic conformational dynamics of the NBDs in surface-immobilized McjD in proteoliposomes.**

A   Representative FRET time traces (black, FRET signal; yellow, fit) of McjD NBD C547 labeled with Alexa 555 and Alexa 647-maleimide under apo conditions at 10 ms time resolution. Approximately 85% of all apo-McjD traces show no significant fluctuations beyond shot noise. Apo-McjD is predominantly in the low FRET state with E* values < 0.3, which indicates a predominant open state.

B, C   Within the complete data set of apo-McjD (*N* = 80 traces) comprising donor and acceptor, ~15% of all molecules show infrequent fluctuations to a higher FRET efficiency state with a lifetime of (C) 82 ± 25 ms where E* values > 0.5 are observed during short dwells. The data set represents a total recording time of 4.6 min with a temporal resolution of 10 ms. To exclude unwanted influence on McjD due to surface immobilization, experimental verification is provided that ATP-induced NBD switching occurs similarly with detergent-solubilized McjD on the surface in Fig EV4.

ATPase activity needs to be less controlled than TMD movements since the TMDs represent the gate toward the extracellular milieu.

**ATP-induced conformational dynamics of the NBDs of McjD**

Finally, we aimed to study ligand-induced movements of McjD (Fig 5). The complexity of the McjD system and its "ligands" MccJ25 and ATP asks for testing of multiple conditions, i.e., equilibrium and non-equilibrium cases. Both peptide transport or AMPPNP-binding would require triggered addition of the compounds; such experiments are best studied with TIRF microscopy in combination with a flow-cell arrangement to quickly exchange buffers and to trigger conformational changes of the transporter at distinct time points (Goudsmits *et al*, 2017a,b). These methods were, however, not available for this study and will be used in a future publication on ligand-induced dynamics of McjD (K. Tassis, M. de Boer, K. Bountra, K. Beis & T. Cordes, unpublished). Confocal scanning microscopy, as used in this study, does not have the parallelized detection and high throughput of wide-field single-molecule methods but better time resolution to detect, e.g., fast events such as intrinsic conformational dynamics (see Fig 4).

Consequently, we decided to study the basal ATPase activity of liposome-reconstituted McjD with our confocal scanning setup and imaged McjD-C547 at concentrations of 120 μM ATP, i.e., concentrations close to the $K_d$ value. From our previously measured

basal catalytic ATPase activity of McjD in liposomes (Choudhury *et al*, 2014) with a $k_{cat}$ of ~3.5 min$^{-1}$, we expect that associated conformational dynamics occur on a timescale of tens of seconds. Consistent with that the experiments show mostly stable FRET traces with either high- or low-FRET values corresponding to the two conformational states of the NBDs that were already described in Fig 3B, since transitions will occur on timescales longer than the average photobleaching lifetime of ~10–20 s. The mix of both types of traces also represents the fact that the protein is half occupied with ATP and half free, either after ATP hydrolysis or prior to ATP binding (Fig 5A). Short-lived traces that have the transporter in the inside-in orientation could not be considered for the data analysis due to their short photobleaching lifetime and the low likelihood to detect a transition.

Strikingly, we find examples where switching between both FRET states occurs either from high to low (Fig 5B) or low to high (Fig 5C) with final photobleaching. These data represent the first direct observation of intrinsic (Fig 4) and ligand-induced conformational dynamics (Fig 5) of an ABC transporter and reveal the relevant timescales of 100 ms and seconds for conformational switching in the NBD domains, respectively. The data in Fig 4 were recorded under active ATP hydrolysis conditions, and we cannot distinguish between occupation of the NBDs by ATP or ADP (as a result of ATP hydrolysis). Our attempts to prepare the ATPase-deficient E506Q mutant in our C547 construct resulted in aggregated protein, thus limiting our in-depth interpretation of these data.

## Discussion and Outlook

From the viewpoint of structures, ABC transporters that utilize the alternating access mechanism are believed to undergo large conformational changes upon ATP binding and hydrolysis (Dawson & Locher, 2006; Ward *et al*, 2007), whereas McjD so far appeared to be more "rigid" (Bountra *et al*, 2017). So far, our only evidence that McjD adopts an outward-open conformation was from transport data using a cross-linked TMD between TMs1-2 and TMs1'-2' (Bountra *et al*, 2017). We have speculated that this conformation might not be sufficiently well populated or long-lived to be probed by EPR techniques. Here, we probed the opening of the McjD TMD by smFRET by placing FRET labels at TM1 and 1'. MsbA adopts the expected outward-open conformation in the presence of nucleotides by movement of TMs1-2 toward the opposite protomer that causes subunit intertwining (Ward *et al*, 2007). Our detergent data revealed that the TMD of McjD does not adopt an outward-open conformation in the presence of nucleotides alone consistent with our previous PELDOR experiments (Fig 2C; Bountra *et al*, 2017). Addition of both ATP and the peptide MccJ25, however, resulted in lower E* values suggesting that TMs1-2 from one protomer have moved toward the opposite protomer for opening of the periplasmic side of the TMD and release of the peptide (Fig 2C). Here, we provide the first direct observation that McjD undergoes conformational changes in its TMD and adopts an outward-open conformation. Strikingly, we also observed similar conformational changes in surface-immobilized McjD in proteoliposomes (Fig 3B). We observe two main populations, one that has a low E* and one with a high E* value suggesting that some McjD molecules have opened up to release the bound peptide, and some are reverting back to the occluded

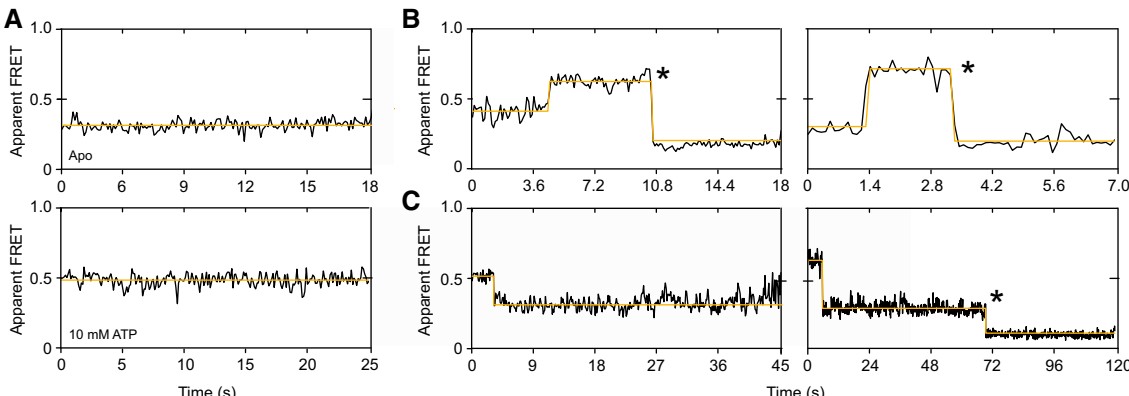

**Figure 5. ATP-induced conformational dynamics of the NBDs in surface-immobilized McjD in proteoliposomes.**

A Representative FRET time traces (black, FRET signal; yellow, fit) of McjD NBD C547 labeled with Alexa 555 and Alexa 647-maleimide at 10 ms time resolution. This panel shows FRET traces of ATP-loaded (bottom) and ATP-free (top) McjD.

B Representative FRET time traces that show switching between ATP-free McjD (E* ~0.3) and ATP-loaded (high FRET > 0.5) at ~4 s (left) and ~1.4 s (right). The traces show photobleaching events of the acceptor after ~10 s (left) and ~3.3 s (right).

C Representative FRET time traces that show switching between ATP-loaded (high FRET > 0.5) and ATP-free McjD (E* ~0.3) at 2 s (left) and 5 s (right). The right trace shows photobleaching after 70 s (acceptor-bleaching).

conformation upon substrate release; we cannot exclude that the population with the high E* values is McjD bound to ATP but not MccJ25. In the ALEX data, we cannot distinguish these individual populations but the histogram analysis shows a rather broad distribution that could be a mixture of similar states. These data suggest that there is a tight coupling between ATP and substrate binding and subsequent release that is significantly different from other ABC transporters. Furthermore, it is the first report to show that both ATP and substrate are required to induce conformational changes in the TMD.

Since ATP binding is coupled to opening of the TMD, we also used a native cysteine, C547, at the NBDs in order to monitor how ATP binding and hydrolysis affect their conformation. The crystal structures of exporters display varying degrees of NBD disengagement in the absence of nucleotides (Ward *et al*, 2007; Lin *et al*, 2015; Perez *et al*, 2015; Bountra *et al*, 2017). These data were questioned as a detergent artifact mistaken for dynamic NBDs. DEER data of ABC transporters in liposomes and bicelles show broad lines for apo NBDs (Dong *et al*, 2005; Borbat *et al*, 2007; Zou *et al*, 2009; Bountra *et al*, 2017), further suggesting that in the absence of nucleotides the NBDs can have different degrees of disengagement. Upon nucleotide binding, the NBDs dimerize and display nearly identical distances across different transporters (Ward *et al*, 2007; Lin *et al*, 2015; Perez *et al*, 2015; Bountra *et al*, 2017). Our smFRET data in detergent and liposomes revealed that the NBDs of McjD adopt very similar disengagement with E* values of 0.3 and larger E* values of 0.5 upon nucleotide binding. These trends are in perfect agreement with our crystal structures (Bountra *et al*, 2017) and suggest that, although the crystal structures have been obtained using detergent-purified protein, they represent relevant conformations of the transport cycle as supported by the direct comparison of detergent and liposome smFRET data.

From structural and biochemical studies, the driving force behind NBD dimerization in the presence of nucleotides remains unclear although molecular simulations have tried to address the dimerization process (Szollosi *et al*, 2018). It has been proposed that

nucleotide binding brings the NBDs closer to each other, but there is no clear consensus on how transporters that display large degree of disengagement can form a closed dimer; binding of substrate at the TMD has been proposed to bring them closer as a result of inducing conformational changes at the TMD upon binding (Ward *et al*, 2007). Our smFRET data from surface-immobilized McjD in proteoliposomes revealed that the NBDs display conformational dynamics in the absence of substrates rather than being static NBDs. So rather than ATP binding alone driving the closure of the NBDs, our data suggest that the apo NBDs can sample the dimerized state without any nucleotide (Fig 4). This is the first direct observation that apo NBDs can sample a dimerized state and it is in agreement with our previous cysteine cross-linking studies at the NBDs that showed cross-linking in the absence of CuCl₂, suggesting a very close proximity to each other (Bountra *et al*, 2017). Unlike the NBDs that show detectable intrinsic conformational fluctuations, the TMD of McjD did not reveal any dynamics in the absence of ligands and even in the presence of nucleotides alone, suggesting that opening of the periplasmic gate is even more tightly coupled to ATP and MccJ25 binding.

In light of the smFRET data and previous structural work, we can provide a very detailed mechanism for the transport of MccJ25 (Fig 6). We propose that, in the absence of substrate MccJ25 (futile ATP hydrolysis), the NBDs infrequently sample different conformations, including a state that is indiscernible from the ATP-closed dimer, which is fully populated in the presence of ATP (Fig 6A). As we have previously shown, the TMD does not open up in the presence of nucleotides alone, and in addition, our smFRET data did not show any dynamics either. On the other hand, in the presence of both ATP and MccJ25 (transport cycle), the TMD of McjD opens as seen directly by smFRET to release the bound peptide and collapses to an occluded conformation that is in agreement with our structural and PELDOR measurements (Bountra *et al*, 2017; Fig 6B).

In conclusion, all our data suggest that ATP and MccJ25 binding is tightly coupled to the opening of the TMD that is distinct from other multidrug ABC transporters. We have shown that McjD can only transport MccJ25 and no other antibacterial peptides or drugs

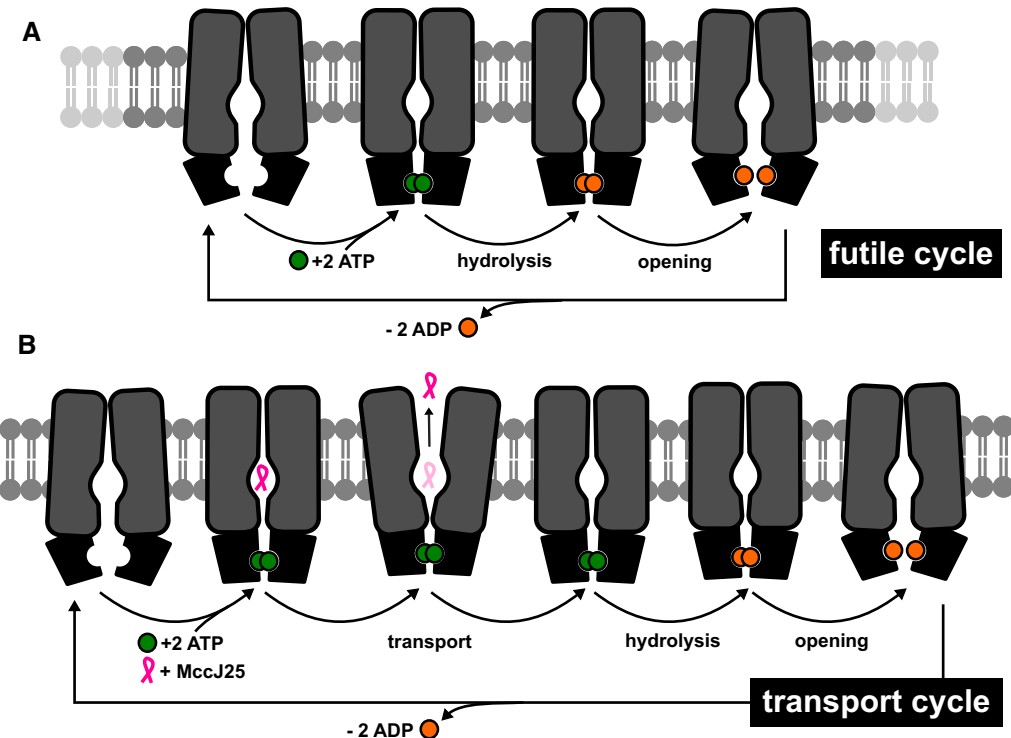

**Figure 6.  Refined transport mechanism of ABC exporter McjD based on structural, biochemical, and biophysical data.**

A, B    Binding of ATP alone (futile cycle) is not coupled to an opening of the TMD that remains occluded. NBDs can close in the presence of ATP and do not require substrate for closure. On the other hand, the NBDs are more dynamic and they can sample a nearly closed dimer that is driven to full closure in the presence of ATP. Binding of both ATP and MccJ25 are coupled to open up the TMD, which reverts back to an occluded conformation upon substrate release. ATP hydrolysis and ADP release reset the transporter to an inward conformation for a new transport cycle.

(Romano *et al*, 2018), and these data confirm that this tight regulation of opening the TMD to the periplasm may be a unique mechanism of natural product self-immunity ABC transporters that distinguishes them from multidrug transporters that only require ATP binding to open their TMD. Therefore, we propose that for multidrug transporters if a ligand can fit in their cavity, they will probably transport it since its binding is not coupled to the opening of the TMD, whereas more specific ABC transporters will only open the cavity in the presence of a single specific substrate.

## Materials and Methods

### Protein expression and purification

The full-length cys-less *mcjD* was subcloned into a pET-28b vector with a non-cleavable His$_8$-tag from the pWaldo-McjD-GFPd vector (Choudhury *et al*, 2014). Single cysteine mutations at positions Y64, L67, and C547 (native cysteine) were introduced using the Quick-Change Site-Directed Mutagenesis Kit (Agilent Technologies). Expression and purification of the McjD variants were performed as described previously (Choudhury *et al*, 2014; Bountra *et al*, 2017). McjD variants for FRET were purified in a final buffer of 20 mM Tris pH 7.5, 150 mM NaCl, and 0.03% (w/w) n-dodecyl β-D-maltoside (DDM). The substrate of McjD, the peptide MccJ25, was produced and purified as previously described (Zirah *et al*, 2011).

### Protein labeling

For labeling, 200 μg McjD or its variants were incubated with 1 mM DTT for 30 min on ice to fully reduce oxidized cysteines. The protein was passed through a Superdex S200 column (GE Healthcare) equilibrated in deoxygenized buffer of 20 mM Tris pH 7.5, 150 mM NaCl, and 0.03% (w/w) DDM. Fractions containing McjD or its variants were pooled and diluted. For labeling, Alexa Fluor 555 C2 maleimide (Thermo Fisher Scientific) and Alexa Fluor 647 C2 maleimide (Thermo Fisher Scientific) were added to McjD at a ratio of 1:4:5 (McjD: Alexa 555: Alexa 647). The mixture was degassed under argon for 5 min followed by gentle shaking for 5 h at 277 K. The labeled protein was passed through a 2-ml Zeba spin desalting column (Thermo Fisher Scientific) equilibrated in the same freshly deoxygenized buffer. Labeled McjD was subsequently applied onto a Superdex S200 column (GE Healthcare). Fractions with optimal labeling efficiency were collected for reconstitution as shown in Fig 1.

### ATPase assays

The ATPase activity of the labeled McjD-L64C, McjD-Y67C, and McjD-C547 was determined as described previously (Choudhury *et al*, 2014). Ligand-induced ATPase activity was determined in the presence of 0.5 mM MccJ25, which was obtained as described before (Choudhury *et al*, 2014).

## Preparation of biotinylated liposomes

Biotinylated lipids were prepared from a synthetic lipid mixture of 67% (w/w) 1,2-dioleoyl-sn-glycero-3-phosphoethanolamine (DOPE), 23% (w/w) 1,2-dioleoyl-sn-glycero-3-phospho-(19-rac-glycerol) (DOPG), 7% (w/w) cardiolipin, and 3% (w/w) 1,2-dioleoyl-sn-glycero-3-phosphoethanolamine-N-(cap biotynil), short biotin-DOPE (Avanti Polar Lipids). Approximately 100 μl of lipid mixture was dissolved at 20 mg/ml in chloroform and dried under vacuum rotary evaporation at 30°C for about 30 min until a dry film was formed. One milliliter diethyl ether was used to dissolve the lipid film, and the lipid mixture was further dried by evaporation. Lipids were then hydrated in a buffer containing 20 mM TrisCl pH 7.5, 150 mM NaCl to a final concentration of 20 mg/ml (w/v). The lipid suspension was sonicated on ice for 16 cycles (15 s on, 45 s off) with an amplitude setting of 70% to generate unilamellar vesicles. Vesicles were flash frozen in liquid nitrogen and subsequently thawed slowly at room temperature. The freeze–thaw process was repeated two times to obtain homogenous product. Prepared liposomes were aliquoted and stored at −80°C.

## Reconstitution of McjD in proteoliposomes

Labeled McjD variants were reconstituted in either polar lipids for ALEX measurements (Fig EV3A) or biotinylated proteoliposomes for surface measurements using the rapid dilution method as previously described (Bountra *et al*, 2017). In brief, *E. coli* polar lipids (20 mg/ml in water) were bath sonicated for 15 min until the solution was less hazy. One hundred microliter of 0.1 mg/ml McjD or the respective mutant were mixed with 20 μl lipid stock on ice for 10 min. The solution was diluted into 4 ml of liposome buffer (50 mM Tris pH 8 and 50 mM KCl) and was incubated for further 5 min on ice. Proteoliposomes were pelleted by centrifugation at 45,000 *g* for 30 min. The supernatant was removed, and the pellet was resuspended in 100 μl liposome buffer. Biotinylated liposomes were sonicated in water bath for 15 min to dissolve the lipid. Labeled McjD was added at a protein-to-lipid ratio of 1:1,000 (w/w). The reaction was followed with incubation on ice for about 5 min, and the mixture was diluted further into 4 ml of liposome buffer and incubated on ice for 5 min. Proteoliposomes were collected by centrifugation at 33,000 × *g* for 30 min and resuspended into 100 μl of liposome buffer to be used in experiments. Since McjD in proteoliposomes proofed to be stable only for a few hours, freshly prepared samples were used for all experiments.

## Transport assay

For transport assays, the amount of protein reconstituted in liposomes was 0.4 mg/ml and the final Hoechst concentration was 0.1 μM (Fig EV3C). Data were recorded using a Cary Eclipse Fluorescence Spectrophotometer (Agilent Technologies).

## Orientation of McjD in proteoliposomes

To check the orientation of McjD in proteoliposomes, wild-type McjD carrying a C-terminal cleavable TEV-His-tag was purified in 0.03% DDM and reconstituted into proteoliposomes as described before (Fig EV3B; Bountra *et al*, 2017). The ratio of McjD to proteoliposomes was the same as for the labeled McjD variants used in smFRET measurements. Detergent-purified McjD-His (10 μg) and liposome-incorporated McjD-His (10 μg) were incubated with TEV protease-His at 298 K for 3 h using a protein-to-TEV molar ratio of 1:1. Samples before and after cleavage were loaded onto an SDS–PAGE gel and analyzed by Western blot using a monoclonal primary antibody (Thermo Fisher) for the McjD-His-tag and the rabbit anti-mouse IgG secondary antibody (Thermo Fisher) according to manufacturer's instructions. The Western blot was imaged using an ImageQuant LAS 4000 (GE Healthcare). The orientation of McjD in liposomes (ratio of McjD-His molecules inside to outside) was calculated by measuring densitometry counts of McjD-His before and after TEV treatment. Densitometry measurements were performed using the ImageQuant TL software (GE Healthcare), and for quantification of band intensities, the automatic peak detection mode was applied. The efficiency of detergent-purified McjD-His cleavage following TEV incubation was also confirmed by densitometry.

## Single-molecule FRET microscopy in solution using alternating laser excitation

Microscope cover slides (no. 1.5H precision cover slides, VWR Marienfeld) were coated with 1 mg/ml BSA for 30 s to prevent fluorophore interactions with the glass material. Excess BSA was subsequently removed with the imaging buffer containing 20 mM Tris–HCl, 150 mM NaCl, 0.03% (w/w) DDM, 1 mM Trolox (photostabilization agent), and 10 mM cysteamine (pH 7.5; Sigma-Aldrich). Labeled McjD was further diluted in imaging buffer to a final concentration of 25 pM. McjD was studied using a custom-built confocal FRET microscope (van der Velde *et al*, 2013) at room temperature. Excitation light at 532 and 640 nm was used in accordance with the fluorophore wavelengths (SuperK Extreme, NKT Photonics, Denmark). Alternation between the two excitation wavelengths was achieved by modulating the light at 50-μs intervals. The output beam was coupled to a single-mode fiber (PM-S405-XP, Thorlabs, United Kingdom) and recollimated (MB06, Q-Optics/Linos, Germany) before entering an oil immersion objective (60×, NA 1.35, UPLSAPO 60XO, Olympus, Germany). Excitation and emission were separated by a dichroic beam splitter (zt532/642rpc, AHF Analysentechnik, Germany) mounted in an inverse microscope body (IX71, Olympus, Germany). Fluorescence emitted by diffusing molecules in solution was collected by the same oil objective, focused onto a 50 μm pinhole and spectrally separated (640DCXR, AHF Analysentechnik, Germany) onto two APDs (τ-spad, < 50 dark counts/s, Picoquant, Germany) with the appropriate spectral filtering (donor channel: HC582/75; acceptor channel: Edge Basic 647LP; both AHF Analysentechnik, Germany).

## Data analysis

Fluorescence photons arriving at the two detection channels (donor detection channel: $D_{em}$; acceptor detection channel: $A_{em}$) were assigned to either donor- or acceptor-based excitation depending on their photon arrival time. The collected photons correspond to donor-based donor emission F(DD), donor-based acceptor emission F(DA), and acceptor-based acceptor emission F(AA). Fluorescent bursts of individual molecules were identified using established procedures. Uncorrected apparent FRET efficiency (E*) represents

the proximity between the two fluorophores and is calculated using the following equation:

$$E^* = \frac{F(DA)}{F(DA) + F(DD)}$$

Stoichiometry (S) is defined as the ratio between the overall green fluorescence intensity over the total green and red fluorescence intensity. Stoichiometry describes the ratio of donor-to-acceptor fluorophores or better their respective brightness ratio in the sample:

$$S = \frac{F(DD) + F(DA)}{F(DD) + F(DA) + F(AA)}$$

One-dimensional E* and S distributions were fitted using a Gaussian function, yielding the mean values of the distribution and the associated standard deviation ($\sigma$).

### Confocal scanning microscopy and data analysis

The general protocol for smFRET analysis was adapted using the methods described previously (Gouridis *et al*, 2015), a detailed description of the setup is provided in Ref. van der Velde *et al* (2013).

For surface immobilization, microscope cover slides (No. 1.5, Marienfeld, Germany) were cleaned by sonication as described before (Gouridis *et al*, 2015) followed by 10-min plasma etching (Plasma Etch, PE-25- JW). Subsequently, functionalization was done with PEG-silane (6–9 PE units, CAS 65994-07-2) and biotin-PEG-silane (MW3400, Laysan Inc., United States) in toluene at 55°C, overnight. After this step, custom-made flow cells were assembled using established procedures (Gouridis *et al*, 2015). To immobilize labeled McjD (or variants) within these flow cells, the surface was incubated for 10 min with a solution containing 0.2 mg/ml neutravidin (Invitrogen, United States) in 20 mM TrisCl pH 7.5 and 150 mM NaCl. Unbound neutravidin was removed by washing with the same buffer. Non-specific binding of labeled McjD to the PEG-surface was excluded by the following procedure. First surfaces were incubated with 50 pM labeled McjD protein in the absence of neutravidin. Only when neutravidin was added before protein incubation, the surface displayed immobilized McjD molecules, while in the absence of neutravidin incubation, there was no McjD immobilization. A typical surface coverage of labeled McjD protein is shown in Fig 3. All experiments were carried out in a degassed buffer under oxygen-free conditions, utilizing an oxygen scavenging system supplemented with 10 mM aged Trolox as a photostabilization agent (Cordes *et al*, 2009).

To observe conformational kinetics of McjD, surface scanning was performed using a XYZ-piezo stage with $100 \times 100 \times 20$ μm range (P-517-3CD with E- 725.3CDA, Physik Instrumente, Germany). The detector signal was registered using a Hydra Harp 400 ps event timer and a module for time-correlated single photon counting (both Picoquant, Germany). Time traces and scanning images were extracted using the custom-made LabVIEW software (Vogelsang *et al*, 2009a,b). Data were recorded with a constant excitation of 532 nm at an intensity of 0.1–0.5 μW (~25–125 W/cm$^2$). Surface scans were shown in false-color representation with orange, green, and red representing double-labeled McjD, McjD with donor fluorophore only, and McjD with acceptor fluorophore only, respectively. A typical 10-μm-sized false-color image is shown in Fig 3.

After each surface scan, the positions of labeled proteins were identified manually; the position information was used to subsequently generate time traces. For this, the confocal excitation spot was centered over the respective molecule to retrieve donor and acceptor fluorescence intensities.

**Expanded View** for this article is available online.

### Acknowledgements

This work was financed by an ERC Starting Grant (ERC-STG 638536 – SM-IMPORT to T.C.) and a Medical Research Council grant (MR/N020103/1 to K.B.). T.C. was further supported by the Center of Nanoscience Munich (CeNS), Deutsche Forschungsgemeinschaft within GRK2062, LMU Excellent, and the Center for Integrated Protein Science Munich (CiPSM). Ki.B. was supported by a Biotechnology and Biological Sciences Research Council (BBSRC) Doctoral Training Partnership (DTP) Studentship. We thank A.K. Cordes for continuous support when writing this manuscript.

### Author contributions

KBe and TC conceived and designed the study, and supervised the project. KBo, MR, and KBe prepared protein samples and performed functional assays. SR prepared the MccJ25 peptide. FH and KT performed single-molecule experiments. FH, KT, and MB analyzed the smFRET data. All authors contributed to discussion of the research and data interpretation. KBe and TC wrote the manuscript with support from all authors.

### Conflict of interest

The authors declare that they have no conflict of interest.

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
