## [Review Process File · The EMBO Journal]

Conformational dynamics of the antibacterial peptide ABC transporter McjD seen by smFRET

Florence Husada, Kiran Bountra, Konstantinos Tassis, Marijn de Boer, Maria Romano, Sylvie Rebuffat, Konstantinos Beis, Thorben Cordes

Review timeline:	Submission date:	20th Jun 2018
	Editorial Decision:	26th Jul 2018
	Revision received:	10th Aug 2018
	Editorial Decision:	31st Aug 2018
	Revision received:	4th Sep 2018
	Accepted:	5th Sep 2018

Editor: Deniz Senyilmaz Tiebe

Transaction Report:

1st Editorial Decision

26th Jul 2018

Thank you for submitting your manuscript entitled 'Conformational dynamics of the antibacterial peptide ABC transporter McjD seen by smFRET' to The EMBO Journal. We have now received two referee reports, which are included below. Given the referees' positive recommendations, I would like to invite you to submit a revised version of the manuscript, addressing the comments of the reviewers. I should add that it is EMBO Journal policy to allow only a single round of revision, and acceptance of your manuscript will therefore depend on the completeness of your responses in this revised version.

Concerning the last paragraph of the referee #2, when I read the manuscript, I did not find the structure confusing per se. Therefore I leave the decision of restructuring the manuscript up to you.

REFEREE REPORTS

Referee #1:

- general summary and opinion

Husada et al. report a single-molecular FRET analysis of the conformational dynamics of the E.coli exporter McjD. They find that in the presence of ATP and the substrate peptide MccJ25 are dynamics associated with formation of an outward-open state observed, whereas in the presence of ATP no major dynamics are observed in the trans-membrane domain (TMD). This is unlike for other and more promiscuous exporter where the ATP cycle alone is associated with full cycle dynamics, and they discuss if this difference could be a characteristic for a substrate-specific exporter like McjD, where tight coupling is perhaps selected for. The findings are addressing important function questions

- Major concerns

Overall the study seems well conducted and using a suitable set of experiments. The following however are major concerns:

- Fig. 1D is used to indicate that the labeled constructs behave more or less like wildtype, but the native C547 labelled enzyme looks however perturbed in activity with higher basal ATPase and relatively less substrate specific activation of the ATPase activity. Combining that with about 60% labelling efficiency it may lead to the question if the labelled enzyme has in fact have lost the substrate coupling entirely? That potentially affects all other discussions of the paper.
- The findings of strict dependence of the transported peptide substrate MccJ25 are not in line with the authors' previous findings that in the absence of MccJ25, McjD transports other substrates, such as Hoechst 33342, i.e. like other drug resistance exporters (Choudhury et al. 2014, PNAS) - i.e. a full functional cycle is obtained without the specific substrate.
-
- minor concerns that should be addressed
- The system environment is described as "native environment" (page 3), which is probably too much of a stretch
- Fig. 1D (again) - how many replicates/duplicates went into the block diagram?
- Fig. 2 A+B perhaps the micelle cartoon is a bit misleading for the untrained reader, covering the entire protein.
- The dynamics of the labelled Y64C are described as non-physiological because the nearby L67C do not show such behavior. That appears like an unreasonable argument. What do electron density maps and MD simulations predict for the native Tyr64 residue? Could it be that this residue position is simply dynamic?

Referee #2:

- general summary and opinion about the principle significance of the study, its questions and findings

The paper by Husada and coworkers presents single molecule FRET measurements of the E. coli transporter protein McjD, which exports an antibacterial peptide. The goal of the paper is to provide evidence to support or dismiss the two leading models of the mechanism of action, alternating access vs. outward-only mechanisms. The dimeric protein is labeled at 3 locations for distinct experiments, two in the outward tip of the transmembrane domains (TMD) and one in the cytosolic internal nucleotide binding domains (NBD). Each individual mutation yields two labels in the dimeric protein, which are labeled randomly by donor and acceptor fluorophores. FRET measurements of these label pairs are performed in both lipid bilayers and detergent solution as well as in the presence/absence of target peptide and ATP, ADP, AMPNP, ATP-vanadate, ADP-vanadate, and Apo (no nucleotide). Population distributions are observed to shift under different conditions and some transitions between FRET states are characterized as well.

The main observations are: that ATP can close the NBD domains, which only reopen after ATP hydrolysis; that ATP+peptide substrate cause opening of the TMDs; under other conditions the TMDs do not move much. In addition, conformational dynamics of the NBDs are observed in the absence of ligand/nucleotide that are very fast (less than 100 msec) and much slower with ATP close to the equilibrium binding concentration of the ATP binding sites (order tens of seconds). These observations are argued to support a transport mechanism that is similar to the alternating access model and that TMD opening is tightly coupled to binding of ATP+target peptide. The general topic of transporter protein mechanism is important for many areas, not the least being anti-bacterial approaches. The method of single molecule FRET is appropriate to address the questions of dynamic conformation this study proposes. I will rely on other reviewers to comment on the significance of the study in the specific area of membrane transport proteins, but I have some concerns about the details of the single molecule FRET results.

- specific major concerns essential to be addressed to support the conclusions

1. Although the authors correctly state they do not rely on the quantitative conversion of FRET to distance, but rather only the relative changes in FRET to define different states, more explanation

should be provided for the difference in absolute FRET values between the lipid and detergent experiments. These were somewhat difficult to reconcile for this reviewer. A table in the supplement listing all the FRET values fitted for all the histograms would be helpful. There is a large difference in FRET for each nucleotide condition between the detergent and the lipid experiments. For example, comparing figure 2 and 3, L76C apo in detergent is FRET 0.95 but in lipid is FRET close to 0.70. This is a dramatic difference and the origin of the change due to detergent vs. lipid should be determined before concluding that these FRET states report the same conformation of the protein. Similar differences are seen for most of the other FRET values between these conditions.

2. Related, the word "identical" in the section heading "McyjD shows identical conformational changes in detergent and liposomes" seems to be too strong. Trends in FRET changes with nucleotide are similar, but given the absolute differences in FRET values, this may be over-reaching to say identical. This whole section may need to be rewritten with this perspective adjusted.

3. The unexplained extra state in the T64C apo data in the lipid experiments is somewhat disturbing. Can some more concrete explanation be discovered?

4. In figure 3A, why is there no 60/40 population split seen in the apo vs. atp data in the lowest panel for C547 experiments like was seen with the other label sites (L67C and Y64C) in the upper panels?

5. The photophysical issues mentioned at the bottom of page 7 are confusing. What is meant by the sentence "The fact that the orientational bias plays a minor role for data evaluation in the ATP-bound state for the NBDs are altered photophysical properties of the dyes."

6. The language should be softened about the closed TMD states not altering their conformational states in the lipid-based set of experiments. It is difficult to claim no changes when fret is this close to 1.0. FRET does not have much sensitivity in these distance ranges.

7. For figure 2 and S1 data, the control experiment for L67C label site of MccJ25 substrate in the absence of ATP seems to be missing. Also, for the C547 label site, it would be interesting to present the data with the MccJ25 substrate.

- minor concerns that should be addressed

1. Figure 1c, mention the column in the caption. It is difficult to search for the methods to find it.

2. Mention what the native cysteine at 547 is mutated to when labeling other cysteine mutations at Y64 or L67

3. Figure 4A caption is not clear describing what the two panels are. The apo vs. atp difference should be mentioned in the caption. The lowest panel in fig 4b also does not mention what it is (apo I guess).

4. Is the first paragraph of the section on "ATP-induced conformational dynamics of the NBDs of McyjD" necessary?

5. In figure 5b left panel, the donor photobleaching at 40 s is not particularly obvious when plotting FRET because the noise is so high after bleaching.

6. In methods, define DDM when first used.

7. In methods, the section about reconstitution of McyjD in proteoliposomes is unclear. More explicit detail about mixed volumes and concentrations is needed to determine lipid, protein and detergent ratios at the steps during reconstitution. Is pelleting and resuspension sufficient to remove detergent or is some residual detergent expected?

8. The orientation section in the methods should explicitly point to supporting figure S3.

9. In supporting figure S3A, there are 10 lanes in the gel (in addition to the markers lane) but there

are only 8 labels across the top. What are the 2 extra plus in the gel? Maybe my layout of that figure was not correct and the plus/minus across the top was jumbled.

10. Figure S5 was confusing. The cartoon suggests it was TMD labels, but they are not specified. Is it 67C or 64C? Maybe not because apo for these sites yields high fret (0.95 or so) in detergent (figure 2). These FRET traces are much lower - near 0.5. What mutant is this figure and display FRET histograms for this detergent immobilized data. It seems different than detergent in solution like figure 2.

- any additional non-essential suggestions for improving the study (which will be at the author's/editor's discretion)

I suggest the authors consider their choice for the structure of the paper. The results in lipid bilayers are much more relevant than the detergent studies. Presenting the detergent results first seems unnecessary and distracting. The paper might be more focused if they exclusively use the lipid studies in the main text and place the detergent studies in a supplement to support the simple point that the trends in detergent solution were similar to the lipid results.

1st Revision - authors' response

10th Aug 2018

We thank you for the evaluation of our submission “**Conformational dynamics of the ABC transporter McjD seen by single-molecule FRET**” by Husada, Bountra and co-workers (EMBOJ-2018-100056). We were very pleased about your invitation to revise the manuscript and attach the revised version of the paper (and requested editorial material), where we followed the recommendations of the two referees. We think the manuscript did greatly benefit from the constructive referee comments. We hope you find the revised version suitable for publication in EMBO Journal.

We provide a point-by-point response to the referee comments below. In summary, we followed all comments of the referees (but refrained from restructuring the detergent/liposome data sets as suggested by referee #2 as a request). For your convenience we summarize the most important changes here:

- (i) We provide additional data regarding the biochemical characterization of labelled McjD (concern of referee #1) in Figure EV3C.
- (ii) We discuss the FRET approach and the use of setup-dependent apparent FRET E^* in more detail and provide tables for mean E^* values as suggested by referee #2. This also triggered us to rewrite the comparison section between conformational states of McjD in detergent and liposomes as suggested by referee #2.
- (iii) We provide the requested control experiment of Mccj25 (substrate) in the absence of nucleotide and integrated the data into new Figure EV1B (referee #2).

Referee #1:

- general summary and opinion

Husada et al. report a single-molecular FRET analysis of the conformational dynamics of the E.coli exporter McjD. They find that in the presence of ATP and the substrate peptide MccJ25 are dynamics associated with formation of an outward-open state observed, whereas in the presence of ATP no major dynamics are observed in the trans-membrane domain (TMD). This is unlike for other and more promiscuous exporter where the ATP cycle alone is associated with full cycle dynamics, and they discuss if this difference could be a characteristic for a substrate-specific exporter like McjD, where tight coupling is perhaps selected for. The findings are addressing important function questions

Our answer: We thank the referee for the summary and this positive appraisal of our work.

- Major concerns

Overall the study seems well conducted and using a suitable set of experiments. The following however are major concerns:

- Fig. 1D is used to indicate that the labeled constructs behave more or less like wildtype, but the native C547 labelled enzyme looks however perturbed in activity with higher basal ATPase and relatively less substrate specific activation of the ATPase activity. Combining that with about 60% labelling efficiency it may lead to the question if the labelled enzyme has in fact have lost the substrate coupling entirely? That potentially affects all other discussions of the paper.

Our answer: We disagree with the referee that labelling has perturbed its activity. We agree that its basal ATPase activity of C547 is marginally higher, but if the transporter had lost its coupling to MccJ25, we would have expected a much higher ligand induced ATPase activity, similar to other multidrug transporters. We provide transport data (Figure EV3C) that show that its transport activity is not affected by the labelling; C547 has similar uptake rates as the wild type McjD, suggesting that ATP and substrate are still tightly coupled.

- The findings of strict dependence of the transported peptide substrate MccJ25 are not in line with the authors' previous findings that in the absence of MccJ25, McjD transports other substrates, such as Hoechst 33342, i.e. like other drug resistance exporters (Choudhury et al. 2014, PNAS) - i.e. a full functional cycle is obtained without the specific substrate.

Our answer: We have recently published a paper on the selectivity of McjD (Romano M, et al, (2018) ACS Chemical Biology, DOI: 10.1021/acscchembio.8b00226) and we have shown that McjD can only recognize MccJ25 and no other peptides or drugs, ie it is not a multidrug transporter. Although, it appears to be specific for MccJ25 only, it can also transport the dye Hoechst due to non-specific interactions, which we commented in the ACS Chem Biol manuscript. Hoechst induces the same conformational changes as with MccJ25 based on cys-crosslinking based transport assays (Bountra K, et al, (2017), DOI:10.15252/embj.201797278).

- minor concerns that should be addressed

- The system environment is described as "native environment" (page 3), which is probably too much of a stretch

Our answer: We agree with the referee here and replaced this term by “native-like lipid environment” throughout the manuscript.

- Fig. 1D (again) - how many replicates/duplicates went into the block diagram?

Our answer: We performed 2 replicates from 2 independent reconstitutions. We have modified the figure legend accordingly.

- Fig. 2 A+B perhaps the micelle cartoon is a bit misleading for the untrained reader, covering the entire protein.

Our answer: We corrected the imprecise representation of the detergent micelle according to the referee's suggestion. We provide a new cartoon that should be more accurate, which is now used in Figure 2 and S6.

- The dynamics of the labelled Y64C are described as non-physiological because the nearby L67C do not show such behavior. That appears like an unreasonable argument. What do electron density maps and MD simulations predict for the native Tyr64 residue? Could it be that this residue position is simply dynamic?

Our answer: We agree with the referee that our initial interpretation of the Y64C dynamics as irrelevant may not have been very rational. Y64C appears to adopt a rather 'rigid' conformation due to its interaction with K48; the two residues are forming a weak hydrogen-bond, 3 Å. Labelling of the Y64C would result in loss of this hydrogen bond, thus making it more prone to flexibility, which is consistent with L67C being more rigid closer to the membrane as we do not observe broad distributions. We can also exclude interaction of the label with the membrane since we do not observe such dynamics with L67C that is closer to the membrane. We have modified our manuscript accordingly.

Referee #2:

- general summary and opinion about the principle significance of the study, its questions and findings

The paper by Husada and coworkers presents single molecule FRET measurements of the E. coli transporter protein McjD, which exports an antibacterial peptide. The goal of the paper is to provide evidence to support or dismiss the two leading models of the mechanism of action, alternating access vs. outward-only mechanisms. The dimeric protein is labeled at 3 locations for distinct experiments, two in the outward tip of the transmembrane domains (TMD) and one in the cytosolic internal nucleotide binding domains (NBD). Each individual mutation yields two labels in the dimeric protein, which are labeled randomly by donor and acceptor fluorophores. FRET measurements of these label pairs are performed in both lipid bilayers and detergent solution as well as in the presence/absence of target peptide and ATP, ADP, AMPNP, ATP-vanadate, ADP-vanadate, and Apo (no nucleotide). Population distributions are observed to shift under different conditions and some transitions between FRET states are characterized as well.

The main observations are: that ATP can close the NBD domains, which only reopen after ATP hydrolysis; that ATP+peptide substrate cause opening of the TMDs; under other conditions the TMDs do not move much. In addition, conformational dynamics of the NBDs are observed in the absence of ligand/nucleotide that are very fast (less than 100 msec) and much slower with ATP close to the equilibrium binding concentration of the ATP binding sites (order tens of seconds). These observations are argued to support a transport mechanism that is similar to the alternating access model and that TMD opening is tightly coupled to binding of ATP+target peptide.

The general topic of transporter protein mechanism is important for many areas, not the least being anti-bacterial approaches. The method of single molecule FRET is appropriate to address the questions of dynamic conformation this study proposes. I will rely on other reviewers to comment on the significance of the study in the specific area of membrane transport proteins, but I have some concerns about the details of the single molecule FRET results.

Our answer: We thank the referee for the summary and this positive appraisal of our work.

- specific major concerns essential to be addressed to support the conclusions

1. Although the authors correctly state they do not rely on the quantitative conversion of FRET to distance, but rather only the relative changes in FRET to define different states, more explanation should be provided for the difference in absolute FRET values between the lipid and detergent experiments. These were somewhat difficult to reconcile for this reviewer. A table in the supplement listing all the FRET values fitted for all the histograms would be helpful. There is a large difference in FRET for each nucleotide condition between the detergent and the lipid experiments. For example, comparing figure 2 and 3, L76C apo in detergent is FRET 0.95 but in lipid is FRET close

to 0.70. This is a dramatic difference and the origin of the change due to detergent vs. lipid should be determined before concluding that these FRET states report the same conformation of the protein. Similar differences are seen for most of the other FRET values between these conditions.

Our answer: We are extremely grateful that the reviewer brought this very important point to our attention. We prepared the requested table of mean FRET values and indeed it proved to be extremely useful for a better description of the results. As can be seen in Table 1/EV2 and Figure EV2 including the related changes in the text parts, the quantitative values are useful for a direct comparison of detergent and liposomes. In light of these, we removed statements that were not supported by our data (e.g., statements claiming identical conformations/changes) and provide now a more balanced discussion of the differences in the text.

2. Related, the word "identical" in the section heading "McjD shows identical conformational changes in detergent and liposomes" seems to be too strong. Trends in FRET changes with nucleotide are similar, but given the absolute differences in FRET values, this may be over-reaching to say identical. This whole section may need to be rewritten with this perspective adjusted.

Our answer: This point is valid and we have adjusted the text as detailed under point 1. We added sentences such as "The observed FRET efficiency values for apo-McjD in proteoliposomes are fully consistent but not identical with our detergent data." which can be seen in our comparison document.

3. The unexplained extra state in the T64C apo data in the lipid experiments is somewhat disturbing. Can some more concrete explanation be discovered?

Our answer: We agree with the referee that this point was insufficiently addressed and rewrote the section to read "We observed a bimodal distribution of the Y64C apo state which we associate to the fact that Y64C is being located further up TM1 and is likely be less restricted in motion by the lipids and therefore more dynamic showing a broader distribution. This will not manifest itself in detergent experiments since here the overall distances are shorter and thus less ideal for the dynamic range of FRET to capture it.". We are aware that this does not clarify the query in full, but we cannot give more details on this aspect at the moment due to lacking experimental support for an in-depth interpretations. Since both TMD mutants consistently report on the same type of movement (as seen in Figure 2/3) and show similar biochemical activity, we are convinced of the validity of the data from both mutants and plan to clarify the details of this problem in a future publication.

4. In figure 3A, why is there no 60/40 population split seen in the apo vs. atp data in the lowest panel for C547 experiments like was seen with the other label sites (L67C and Y64C) in the upper panels?

5. The photophysical issues mentioned at the bottom of page 7 are confusing. What is meant by the sentence "The fact that the orientational bias plays a minor role for data evaluation in the ATP-bound state for the NBDs are altered photophysical properties of the dyes."

Our answer: We are grateful for the referee to bring up this point, which we apparently did not properly discuss in the original version of the manuscript. We added the following discussion to make clear where this apparent discrepancy originates from: "Importantly, we found that the random orientation of McjD in liposomes is less clearly visible in Figure 3B for C547 compared to both TMD mutants. We found the origin of this in altered photophysical properties of the dyes. For C547, there is a significant fraction of molecules (30-50%) that display very short photobleaching lifetimes <1 s, in contrast to long-lived molecules similar to those shown in Figure 3B. All short-lived molecules show low-FRET (apo) character independently of presence or absence of ATP. This tells us that whenever the transporter is labelled at the NBDs it requires photostabilizer in the buffer for

long traces. When the fluorophores residue on the TMDs, however, these intrinsically promote higher photostability and more uniform traces from both transporter orientations are observed. As a consequence in the TMD case both populations (inside-out and outside-in) are equally weighted in the cumulative FRET-histogrammes, while for C547 the photostable (NBD out) population dominates the cumulative histogrammes over the shorter traces from the NBD in orientation that cannot react on addition of ATP.”

6. The language should be softened about the closed TMD states not altering their conformational states in the lipid-based set of experiments. It is difficult to claim no changes when fret is this close to 1.0. FRET does not have much sensitivity in these distance ranges.

Our answer: We agree with the referee that we have to be more careful about such statements. In light of the improvements available via both Tables 1/EV1 and associated changes to the text (point 1-3), we believe this issue is resolved.

7. For figure 2 and S1 data, the control experiment for L67C label site of MccJ25 substrate in the absence of ATP seems to be missing. Also, for the C547 label site, it would be interesting to present the data with the MccJ25 substrate.

Our answer: We agree with the referee that these data were missing and inserted them in the form of new Figure EV2B. MccJ25 has no effect on the NBD side and smaller effects on the TMD side compared to ATP+MccJ25 for L67C (see Table 1/EV1) and similar for Y64C. Our previous reason to omit them from the paper was to avoid confusion about the fact that MccJ25 causes small alterations of TMD states, which is however related to rebinding (not transport/alternating access) as also described in (Grossmann N, et al, (2014), Nat Comms, 5419). We have also modified the manuscript to account for this observation and agree with the referee that it should be included for completeness.

- minor concerns that should be addressed

1. Figure 1c, mention the column in the caption. It is difficult to search for the methods to find it.

Our answer: We conducted the suggested change.

2. Mention what the native cysteine at 547 is mutated to when labeling other cysteine mutations at Y64 or L67

Our answer: We conducted the suggested change and inserted this detail into the text (change of cysteine to serine).

3. Figure 4A caption is not clear describing what the two panels are. The apo vs. atp difference should be mentioned in the caption. The lowest panel in fig 4b also does not mention what it is (apo I guess).

Our answer: We conducted the suggested change.

4. Is the first paragraph of the section on "ATP-induced conformational dynamics of the NBDs of McjD" necessary?

Our answer: We consider the paragraph essential since it provides a balanced discussion over our experimental strategy and our future strategy, e.g., an outlook. We would prefer to keep it.

5. In figure 5b left panel, the donor photobleaching at 40 s is not particularly obvious when plotting FRET because the noise is so high after bleaching.

Our answer: We modified Figure 5 as a whole to distinguish between photobleaching and FRET changes easier; now the apo/ATP-bound state is easier compared to the steps shown in Figure 5.

6. In methods, define DDM when first used.

Our answer: We conducted the suggested change.

7. In methods, the section about reconstitution of McjD in proteoliposomes is unclear. More explicit detail about mixed volumes and concentrations is needed to determine lipid, protein and detergent ratios at the steps during reconstitution. Is pelleting and resuspension sufficient to remove detergent or is some residual detergent expected?

Our answer: We have expanded our Materials and Methods section to describe the reconstitution protocol in some more detail, although, we had referenced our previously published protocol (Bountra K, et al, (2017), DOI: 10.15252/embj.201797278). The rapid dilution method, dilutes the detergent nearly 400-fold that would bring the DDM concentration below its CMC. The final pellet is resuspended in 100ul of buffer that would dilute any residual detergent to a further 100-fold. Therefore, it is unlikely to have the effect of detergent in our measurements.

8. The orientation section in the methods should explicitly point to supporting figure S3.

Our answer: We conducted the suggested change.

9. In supporting figure S3A, there are 10 lanes in the gel (in addition to the markers lane) but there are only 8 labels across the top. What are the 2 extra plus in the gel? Maybe my layout of that figure was not correct and the plus/minus across the top was jumbled.

Our answer: We would like to thank the Referee for bringing this to our attention. Yes, the original file seems to have been corrupted during conversion to pdf. The two additional plus represented the last to two-lanes. This has now been corrected and shown in Figure EV3B.

10. Figure S5 was confusing. The cartoon suggests it was TMD labels, but they are not specified. Is it 67C or 64C? Maybe not because apo for these sites yields high fret (0.95 or so) in detergent (figure 2). These FRET traces are much lower - near 0.5. What mutant is this figure and display FRET histograms for this detergent immobilized data. It seems different than detergent in solution like figure 2.

Our answer: We apologize for this mistake in the cartoon (which shows data of the C547-McjD variant in detergent); the values match the detergent values for C547 in solution (Figure 2). We corrected the Figure, which has the new number EV4. We changed the text accordingly: "We also verified ATP-induced NBD switching with detergent solubilized McjD on the surface to exclude unwanted influence on McjD due to surface-immobilization. Surface-immobilized C547C and free-diffusing molecules show a high degree of consistency (compare Figure 2 with Figure EV4).".

- any additional non-essential suggestions for improving the study (which will be at the author's/editor's discretion)

I suggest the authors consider their choice for the structure of the paper. The results in lipid bilayers are much more relevant than the detergent studies. Presenting the detergent results first seems

unnecessary and distracting. The paper might be more focused if they exclusively use the lipid studies in the main text and place the detergent studies in a supplement to support the simple point that the trends in detergent solution were similar to the lipid results.

Our answer: In agreement with the editor, we decided to not follow this suggestion and left the structure of the paper as it is. We believe that the high consistency between liposomes and detergent data should allow to present both data sets side by side in the main text.

2nd Editorial Decision

31st Aug 2018

Thank you for submitting a revised version of your manuscript. It has now been seen by both of the original referees whose comments are shown below.

As you will see, both referees find that all criticisms have been sufficiently addressed and recommend the manuscript for publication. However, before I can send the official acceptance letter, there are a few editorial issues concerning text and figures that I need you to address.

REFEREE REPORTS:

Referee #1:

The authors have responded generally well to the comments and requests and provide valuable new data. Reviewer 2 raised some very important points on the absolute FRET values questioning the identity of the lipid and detergent form. Reviewer 2 suggested that the manuscript was focused on the lipid results. The authors prefer to maintain the structure of their manuscript; in that case I think they must respond better to the structural difference between the detergent form and the lipid form - what may it result from?

Referee #2:

I am satisfied with the authors' responses and the revisions in the paper. I have no other critical comments.

2nd Revision - authors' response

4th Sep 2018

We have addressed all the Editorial queries and we have uploaded the revised manuscript and files.

We have also addressed Referee's #1 comment about the differences between the detergent and lipid FRET values.

Referee #1:

The authors have responded generally well to the comments and requests and provide valuable new data. Reviewer 2 raised some very important points on the absolute FRET values questioning the identity of the lipid and detergent form. Reviewer 2 suggested that the manuscript was focused on the lipid results. The authors prefer to maintain the structure of their manuscript; in that case I think they must respond better to the structural difference between the detergent form and the lipid form - what may it result from?

We have added the following clarification in p7 of the revised manuscript:

‘Altogether, the combined data of detergent-solubilized and liposome-reconstituted McjD are not identical but consistent – a finding that is interesting in itself; this observation is consistent with previous reports that MsbA displays different degrees of domain separation in micelles and lipid

bilayers (Zoghbi et al, 2016). We have shown that lipids modulate the activity of McjD (Mehmood et al, 2016) and the non-identical FRET values in this study between the detergent and liposome data provide further evidence on the role of lipids on modulating the transport activity of ABC transporters.'

Corresponding Author Name: Konstantinos Beis, Thorben

Journal Submitted to: The EMBO Journal

Manuscript Number: EMBOJ-2018-100056